# Physicochemical properties of stingless bees (*Meliponula beccarii*) honey in Dandi and Meta Robi districts of West Shewa zone, Ethiopia

Desalegn Begna[1]*, Girma Motuma[2◉], Shambel Boki[3◉], Niguse Bekele[4‡], Tamiru Kuru[4‡], Achalu Chimdi[5‡]

**1** Agriculture and Rural Development Center, Policy Institute Study Institute, Addis Ababa, Ethiopia, **2** Department of Animal Science, College of Agriculture and Veterinary Science, Ambo University, Mamo Mezemir Campus, Guder, Ethiopia, **3** Department of Animal Science in the College of Agriculture and Veterinary Science, Ambo University, Mamo Mezemir Campus, Guder, Ethiopia, **4** Department of Natural Resource Management, Ambo University, Mamo Mezemir Campus, Guder, Ethiopia, **5** Department of Soil Sciences, Ambo University, Mamo Mezemir Campus, Guder, Ethiopia

◉ These authors contributed equally to this work.
‡ NB, TK and AC also contributed equally to this work
* dbegna67@gmail.com

**Data Availability Statement:** All relevant data is presented in the manuscript. However, additional row data in Excel has been uploaded as

## Abstract

The study was conducted to characterize the physicochemical properties of honey produced from underground nesting stingless bees (*Meliponula beccarii*) in the Dandi and Meta Robi districts of the West Shewa zone, Ethiopia. A total of 27 honey samples, including 24 samples collected through careful investigation and excavation of natural nests and 3 samples purchased from the market, were analyzed in the laboratory. The evaluated physicochemical properties showed an overall mean of 306.64±87.95 meq./kg free acidity, 28.05±3.52% moisture content, 1.31±0.44 mS/cm electrical conductivity, 3.29±0.16 pH, 0.89±1.14 mg/kg HMF, 0.63±0.24% mineral (ash), 9.39±4.26% glucose, 0.24±0.01 g/100g sucrose, 10.81±4.95 g/100g maltose, and 16.57±2.55 g/100g fructose, turanose (0.20 ± 0.00 g/100g). The differences between the two district's honey samples were significant (p > 0.05) for fructose value and considerable for free acidity, moisture content, and pH values. The honey samples purchased from the market showed similar physicochemical properties to the honey from the feral nests, with a mean of 314.33±88.72 meq./kg free acidity, 27.73±2.52% moisture content, 1.43±0.41 mS/cm electrical conductivity, 3.26±0.13 pH, 0.95±1.23 mg/kg HMF, 0.59±0.19% mineral (ash), 10.11±4.11% glucose, 0.25±0.02 g/100g sucrose, 11.23±4.52 g/100g maltose, and 16.33±2.41 g/100g fructose. The study found that the stingless bee honey from the study areas had distinctive low HMF, high free acidity, and low pH values, which may indicate the honey's potential medicinal properties. The high free acidity in the *Meliponula beccarii* honey appear unusually elevated compared to other stingless bee honey and the Codex Alimentarius standards for Apis honey, suggesting increased fermentation that can originate from the bee species, plant and geographical origins, improper handling, and high moisture content. This study demonstrated that the honey in the study areas has distinctive physicochemical properties from A. mellifera-produced honey, which may support its traditional medicinal uses.

supplemental information. This time, all personal information has been removed or anonymized as per the editor's comment.

**Funding:** The author(s) received no specific funding for this work.

**Competing interests:** The authors have declared that no competing interests exist.

Further detailed studies on ground-nesting and other stingless species' honey medicinal values are recommended to provide scientific evidence.

## Introduction

Honey is a sweet, complex substance produced by bees through a series of processes before being stored in honeycombs or pots until it ripens [1]. It is primarily composed of sugars, mainly monosaccharides [2–4]. Stingless bees, unlike their honeybee (*Apis Mellifera*) counterparts, produce honey from the nectar of flowering plants and store it in pot-shaped structures made of wax cerumen [5].

Studies have shown that stingless bee honey and *Apis* honey possess distinct physicochemical properties. Stingless bee honey generally exhibits greater moisture content, higher acidity, lower sugar composition, and lower enzyme activity compared to Apis honey [6, 7]. The chemical composition of stingless honey can also vary based on factors such as floral origin [6, 8, 9], geographical origin, seasonal and environmental conditions, and handling techniques [10–13].

Ethiopia is home to diverse stingless bee species, including the ground-nesting *Meliponula beccarii*, whose honey, locally known as "*damma daamu*," is commonly harvested and used as traditional medicine [3, 14, 15]. *Meliponula beccarii*, a stingless bee species, is commonly found in Ethiopia and is well known for its honey production as well as its contributions to local biodiversity and pollination. It is particularly prevalent in the Dandi and Meta Robi districts of the West Shewa zone, where it plays a vital role in the ecosystem [16–18]. However, the destructive harvesting [3] methods employed may endanger the native stingless bee species and compromise the honey quality [3]. Characterization of the physicochemical properties of stingless bee honey in some districts of West Shewa like Cheliya, Jeldu, Wolmera, and Tokekutaye has been studied [3]. The Dandi and Meta Robi districts of Ethiopia are known for their stingless bee honey production and traditional medicinal uses. Despite this, the quality and authenticity of the stingless bee honey reported from other districts of west Shewa, information is lacking for Dandi and Meta Robi districts. Therefore, the primary purpose of this study is to evaluate the physicochemical properties of honey from the native stingless bee (*Meliponula beccarii*) in these two districts.

## Material and methods

### The study areas

Dandi and Meta Robi are districts located in the West Shewa Zone of the Oromia Regional State in Ethiopia. These two districts were purposefully selected from the West Shewa Zone due to their high potential for stingless bee honey production and ease of access during data collection. According to the 2021/22 Annual Reports from the Agricultural Offices, Dandi district is located at 8°54'0.11"N latitude and 38°25'0.35"E longitude, and experiences minimum and maximum temperatures ranging from 9.3°C to 23.8°C, with annual rainfall between 750–1,300 mm. Meta Robi district is located at 9°19'0.6"N latitude and 38°09'0.6"E longitude and experiences minimum and maximum temperatures ranging from 20°C to 30°C, with annual rainfall between 750–1,300 mm). The locations were purposefully chosen for the study representing the high potential areas of stingless bee, *Meliponula beccarii*, and with diverse floral composition.

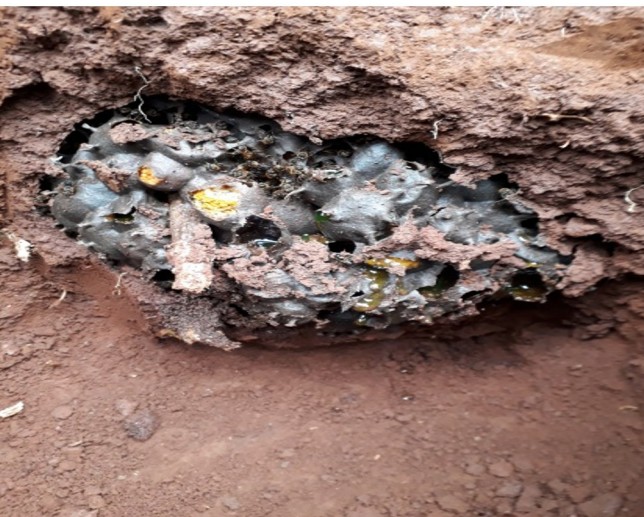

**Fig 1. Excavated feral stingless bee nest.**

## Sample acquisitions

Careful investigation and excavations of 24 wild stingless bee nests (*Meliponula beccarii*) (8 from Dandi district and 16 from the Meta Robi district) were conducted. From each nest, fresh honey samples ranging from 0.25 to 0.5 liters were collected using disposable 50 ml syringes. In addition, three honey samples were purchased from the market to be used for comparison, making the total honey samples 27. The collected honey samples were then stored in separate, airtight, and labeled jars (Fig 1) and properly refrigerated at 4°C until the dates of the analysis.

## Methodologies and protocols of the honey analysis

The analysis of the collected honey samples was done at Ethiopian Conformity Assessment Enterprise (ECAE) [19, 20] in Addis Ababa in accordance of the codex Alimentarius commission of the joint FAO/WHO food standards program [21]. The analyzed parameters included sugar content, moisture content, free acidity, pH, electrical conductivity, ash, and HMF each with three replications. Determination of sugars was performed with high-performance liquid chromatography (HPLC) equipped with a differential refractive index (DRI) detector [22] with the separation using $NH_2$ column (4.6 x 250mm, ZORBAX $NH_2$) and a particle size diameter of 5μm with the column was kept at 30°C throughout the analysis. The mobile phase composition was 70% acetonitrile in water. The injection volumes of the samples were 10μl, with a flow rate of 1.3ml/min. Retention times were identified using standards of peak. The honey samples were also spiked with standards in order to verify the identity of the chromatographic peaks. Triplicate injections were performed and for the peak quantification average peak areas were used. Monosaccharaides (glucose, fructose) and disaccharides (sucrose, maltose, turanose) were used as standards to determine the sugar content of the honey samples. A 5g of honey sample was dissolved with 25% methanol in water solution. The solution was transferred to a 100ml volumetric flask. Solution was filtered through 0.45μm syringe and injected to HPLC. A standard curve was used to determine sugar content (fructose, glucose, sucrose, turanose and maltose) and the values were calculated as gram/100gram of honey sample.

A digital refractometer (Abbe refractometer, Leica Mark II Plus) thermo stated at 20 ˚C with the refractive index and regularly calibrated with distilled water was used [23] to determine the moisture content.

The honey samples were homogenized and the surface of the prism was covered evenly with the sample. After 2 minutes, the refractive index was read with refractometer. Each honey samples were measured triplicate and the average value was taken [23]. Free acidity and pH measurement according to [23]. A pH meter was used to measure the pH of a 10% (w/v) solution of honey prepared in deionized water. A 10g honey sample was dissolved in 75 ml of carbon dioxide-free water. It was stirred with the magnetic stirrer, then pH electrode immersed in the solution and the volume of titrate was recorded when titrated with 0.1M NaOH to pH 8.30. The reading was recorded and the free acidity was calculated using the following formula.

$$Free\ acidity (\text{milliequvalent}) = mL\ 0.1M\ NaOH * 10$$

Electrical conductivity was measured using a conductivity meter (HANNA, HI 2550, EC/TDS/NaCl meter). 20% (w/v) honey solution was suspended in deionized water. Potassium Chloride solution (0.1M) was used for the cell constant. The electrical conductance of this solution was read in mS after the temperature has been equilibrated to 20˚C (International Honey Commission, 2009). 20g anhydrous honey was dissolved in distilled water and quantitatively transferred to a 100ml volumetric flask. The electrical conductivity of the honey solution was calculated using the following formula:

$$SH = K * G$$

Where:

SH = electrical conductivity of the honey solution in mS.cm⁻1

K = cell constant in cm⁻1

G = conductance in mS

For temperatures above 20˚C: subtract 3.2% of the value per ˚C

For temperatures below 20˚C: add 3.2% of the value per ˚C

Ash content was determined according to the methods of International Honey Commission (2009). Accordingly, 5 g of honey was placed in combustion pots, which required preheating to darkness. Then, the samples incinerated at high temperature (550˚C) in a burning muffle (THERMO CONCEPT, KLS 45/13, Germany) for 1 h. After cooling at room temperature, the obtained ash weighed and the proportion of ash (Ac) in g/100g honey was calculated using the following formula: -

$$AC = \frac{(m1 - m2)}{m3} * 100$$

Where:

$m_3$ = weight of honey

$m_1$ = weight of dish + ash,

$m_2$ = weight of dish

Hydroxymethylfurfural was identified as Standard solution: 5-(hydroxymethyl-) furan-2-carbaldehyde (HMF) 1, 2, 5, 7 and 10 mg /L aqueous solution was prepared. The absorbance A of the prepared standard solution was determined using an UV spectrophotometer (CECIL, CE 7500, 7000 series) at 285 nm in 1 cm quartz cells with water in the blank cell. The concentration of the standard solutions was calculated from the literature values for molar absorptivity,

$\varepsilon = 16830$ or absorptivity.

$$\text{Concentration}\left(\frac{\text{mg}}{\text{L}}\right) = \frac{A}{1 * 133.57} * 1000$$

Where: A is the absorbance of the standard solution

HMF content was determined with HMIHC (International Honey Commission, 2009) with some modifications. A high-performance liquid chromatography was used with an absorbance scale 285nm using DAD detector. 10g of prepared honey sample was dissolved with 50ml deionized water. The solution was filtered through a 0.45μm syringe filter and read on a HPLC (Agilent, 1260infinity) using a C18 column (3.0 x 250mm) (ZORBAX SB-C18) kept at 30°C throughout the analysis with a particle size diameter of 5μm. The mobile phase composition was 90% deionized water (1% acetic acid) with methanol. The injection volumes of the samples were 10μl, with a flow rate of 0.7ml/min (International Honey Commission, 2009). The HMF content of the samples was calculated by comparing the corresponding peak areas of the samples and those of the standard solutions taking into account the dilution. The collected data were systematically organized, coded, tabulated, and analyzed using the Statistical Package for Social Science (SPSS) software version 20. One-way ANOVA was employed to compute the variations between the means of each variable in every studied parameter and 0.05% tests were considered significance.

## Results and discussion

### Free acidity

In this study, the overall mean free acidity values of the stingless bee honey samples were 306.64 ± 87.95 meq/kg, ranging from 200–424.74 meq/kg (mean 338.79 ± 72.46) for the Dandi district, and 188.25–456.30 meq/kg (mean 274.49 ± 93.10) for the Meta Robi district. The mean free acid value for the purchased honey sample was 348.5 ± 12.0 meq/kg, and there was no significant difference (P>0.05) between the freshly collected honey samples from the two districts and the purchased sample (Table 1).

The mean free acidity values in this study were well above the acceptable limit of international standards for *Apis mellifera* honey (<50 meq/kg) [21], further confirming the acidic nature of the stingless bee honey [6]. Reports from Ethiopia indicate mean free acidity levels of 92.39 ± 4.45 meq/kg [18] and 7.3 ± 0.36 meq/kg [3] for stingless bee honey samples from various parts of the country, both of which are significantly lower than our findings. Our value is also notably higher than the free acidity of stingless bee honey from Southeast Asia (40 to 60 meq/kg) and South America (50 to 70 meq/kg). The elevated free acidity of *Meliponula beccarii* honey in our samples is likely attributed to a combination of diverse floral sources with high organic acid content [24], specific environmental conditions that promote fermentation [25], and unique soil characteristics influencing nectar composition [26], enzymatic activity and low pH content [27].

Honey free acidity is an important quality parameter, as it indicates the presence of organic acids, which contribute to the flavor and aroma of honey [11]. Acidity values can also be useful for discriminating honeys of different floral origins, and honey adulterated with sugar syrup typically has very low free acidity (<1) [28]. The acidity levels in stingless bee honeys can vary significantly based on factors like bee species, geographical origin, and other variables. The free acidity values were reported 26.5 to 66.4 meq/kg for *Melipona scutellaris* honey (Souza et al., 2006), 29.7 ± 4.4 meq/kg for *Plebeia saiqui* honey, and 42.7 ± 4.2 meq/kg for *Tetragonisca angustula* honey (Vit et al., 1998), and 22.4 to 34.6 mEq/kg for *Melipona beecheii* honey (Ruiz-Argueso and Rodriguez-Navarro, 1973). The *Meliponula beccarii* honey studied appears to have

**Table 1. Mean physicochemical properties of stingless bee honey samples (N = 27).**

| Values | Parameters | | | | | | | | | | |
|---|---|---|---|---|---|---|---|---|---|---|---|
| | Free Acidity meq/kg | MC % | EC mS/cm | HMF Mg/kg | pH | Ash (g) | Glu (g) | suc (g) | Malt (g) | Fruc (g) | Tura (g) |
| Dandi district (N = 8) | | | | | | | | | | | |
| Min. | 200.00 | 25.00 | 0.72 | 0.07 | 3.00 | 0.40 | 1.20 | 0.20 | 6.50 | 12.00 | 0.19 |
| Max. | 424.74 | 36.75 | 1.99 | 4.20 | 3.42 | 1.00 | 15.80 | 0.25 | 16.80 | 16.80 | 0.20 |
| Mean± std | 338.79±72.46 | 29.34 ±4.04 | 1.38 ±0.48 | 1.14 ±1.53 | 3.24 ±0.13 | 0.65 ±0.25 | 8.84 ±5.29 | 0.24 ±0.01 | 10.27 ±3.98 | 14.50 ±1.54 | 0.20±0 |
| Meta Robi district(N = 16) | | | | | | | | | | | |
| Min. | 188.25 | 23.45 | 0.75 | 0.06 | 3.00 | 0.36 | 5.80 | 0.22 | 5.20 | 15.90 | 0.19 |
| Max. | 456.30 | 30.80 | 1.85 | 1.30 | 3.60 | 0.95 | 13.40 | 0.26 | 20.00 | 20.00 | 0.20 |
| Mean± std | 274.49± 93.1 | 26.77 ±2.49 | 1.24 ±0.4 | 0.64 ±0.46 | 3.35 ±0.18 | 0.61 ±0.23 | 9.94 ±3.05 | 0.24 ±0.01 | 11.35 ±5.91 | 18.63 ±1.38 | 0.20 ±0.0 |
| Total (Dendi and meta Robi) | | | | | | | | | | | |
| Min. | 188.25 | 23.45 | 0.72 | 0.06 | 3.00 | 0.36 | 1.20 | 0.20 | 5.20 | 12.00 | 0.19 |
| Max. | 456.30 | 36.75 | 1.99 | 4.20 | 3.60 | 1.00 | 15.80 | 0.26 | 20.00 | 20.00 | 0.20 |
| Mean ± std | 306.64±87.95 | 28.05 ±3.52 | 1.31 ±0.44 | 0.89 ±1.14 | 3.29 ±0.16 | 0.63 ±0.24 | 9.39 ±4.26 | 0.24 ±0.01 | 10.81 ±4.95 | 16.57 ±2.55 | 0.20 ±0.0 |
| P-Value | 0.07 | 0.07 | 0.44 | 0.29 | 0.08 | 0.71 | 0.54 | 0.89 | 0.61 | 0.00 | 0.56 |
| Purchased honey (N = 3) | | | | | | | | | | | |
| Mean± std | 348.5±12.08 | 35.40 ±0.79 | 1.50 ±0.14 | 0.28±0.2 | 3.31 ±0.02 | 0.64 ±0.09 | 9.53 ±0.52 | 0.25 ±0.0 | 7.8±2.0 | 13±0.23 | 0.20 ±0.0 |
| Max. | 372.66 | 36.90 | 1.79 | 0.70 | 3.35 | 0.83 | 10.2 | 0.25 | 12 | 13.4 | 0.20 |
| Min. | 336.5 | 34.20 | 1.34 | 0.08 | 3.27 | 0.54 | 8.50 | 0.25 | 5.5 | 12.6 | 0.20 |
| P-Value | 0.078 | 0.001 | 0.005 | 0.01 | 0.30 | 0.005 | 0.20 | 0.03 | 0.01 | 0.01 | |
| Codex Alimentarius Standard for Honey (CODEX STAN 12–1981, Rev.2-2001) | 50 | 20 | 0.8 | 40 | 3.4–6.1 | <0.6 | N/A | 5 | N/A | N/A | N/A |
| Southeast Asia | 40–60 | 30–35 | 0.5–1.0 | 5–10 | 3.5–4.5 | 0.1–0.3 | 25–35 | 1–2 | 1–2 | 35–45 | |
| South America | 50–70 | 28–30 | 0.3–0.8 | 10–20 | 3.0–4.0 | 0.2–0.5 | 25–30 | 1–3 | 1–2 | 40–50 | |

N = Number of samples, MC = Moisture content, HMF = hydroxyl methyl furfural, EC = Electric conductivity, pH = pH value, Std.D = Standard deviation, Min = minimum, Max = Maximum, MC (Moisture content), HMF (hydroxymethylfurfural,) EC (Electrical Conductivity), Tura (Turanose,) Glu (Glucose), Suc (sucrose), Fruc (Fructose), Malt (Maltose)

unusually high free acidity and low pH compared to stingless bee honey from Brazil (Melipona scutellaris), Venezuela (Plebeia saiqui, Tetragonisca angustula), and Mexico (Melipona beecheii), which typically range from 22.4 to 66.4 meq/kg. This suggests the high free acidity may be due to increased fermentation, potentially stemming from factors like differences in bee species and geographical origin [29], improper handling, or high moisture content [30], which requires further investigation. The higher free acidity in honey is directly linked to its enhanced medicinal potential, and honey with higher free acidity (lower pH) exhibits stronger antimicrobial activity against pathogens [31, 32], greater antioxidant capacity [7, 10], improved anti-inflammatory and wound healing properties [33], and effective antibiofilm activity that can disrupt and prevent microbial biofilms [34, 35], collectively highlighting the direct relationship between the high free acidity in honey and its potential for medicinal and nutritional valueof honey.

## Moisture content

Honey moisture content is an important quality parameter due to its impact on fermentation. The overall moisture content of the honey samples was reported to be between 23.45% and

36.75%.(mean 28.05±3.52%). The recorded moisture contents ranged from 25–36.75% (mean 29.34±4.02) for the Dandi district and 23.45–30.80% (mean 26.77±2.49%) for the Meta Robi district. The purchased honey samples had a moisture content of 34.20–36.90% (mean 35.40 ±0.79%), and there was no statistical difference (P > 0.05) between the honey samples from the two districts (Table 1). The moisture content of freshly collected honey was lower than the in marketed honey and this lower moisture content helps prevent fermentation, indicating better quality preservation of the freshly collected honey sample [25].

This finding is consistent with previous studies [3] reported a moisture content of 25.1– 35.0% (mean 29.6±1.4%) for honey samples of the same stingless bee species collected from four districts of Ethiopia [36] also reported similar results. Studies have shown that stingless bee honey from Southeast Asia (Malaysia) typically has a moisture content ranging from 30% to 35% [26] and in contrast, honey from South America has been observed to have slightly lower moisture content, between 28% and 30% and Brazilian stingless bee honey exhibited moisture levels around 28% [24].

Variations in moisture content among honey samples can be affected by factors such as adulteration, humidity, floral origin, soil type, collection period, and processing aspects [12, 37–39]. The moisture contents noted in this study are lower than what was reported for stingless bee honeys from Mexico, Guatemala, and Venezuela (31.4% to 38.74%) [13], but higher than the 13.86±1.06% reported for Melipona species in Nigeria [40].

The standard moisture content limit for *Apis mellifera* honey is 20% or less, as per the Ethiopian [26] and European [23] standards.

## The pH value

The mean pH values ranged from 3.0–3.42 (mean 3.24±0.13) for the Dandi district and 3.0– 3.60 (mean 3.35±0.18) for the Meta Robi district. The purchased honey samples had a pH range of 3.27–3.35 (mean 3.31±0.023). The pH parameter showed no statistically significant differences (p>0.05) among all the sampled honeys (Table 1).

All honey samples showed low pH values, which is characteristic of stingless bee honeys [5, 6]. This finding is consistent with previous studies [3] reported an overall pH range of 3.4–3.9 (mean 3.7±0.15) for stingless bee honey from four districts in Ethiopia [40] reported pH values of 3.75 and 4.21 for stingless bee honey (Melipona species) in Nigeria, and [41] presented similar findings for the genus *Melipona* in Brazil [24]. The pH of honey from Southeast Asia typically varies from 3.5 to 4.5, as reported by [26] and honey from South America exhibits a pH range of 3.0 to 4.0 according to findings by [24], indicating the *Meliponula beccarii* honey from Ethiopia has a pH, which falls within the lower spectrum of both Southeast Asian and South American honeys. Lower pH may enhance the honey's preservation and antibacterial properties, contributing to its quality [25].

The standard pH value for *Apis mellifera* honey is 3.42–6.10 [42, 43]. The low pH value of Melipona honey has great significance for its extraction and storage, as it inhibits the occurrence and development of microorganisms [44, 45] and reduces some infectious diseases [46, 47].

## Ash contents

In this study, the ash contents ranged from 0.40–1.0 (mean 0.65 ± 0.25g) for the Dandi district, 0.36-1g (mean 0.63±0.24g) for the Meta Robi district, and 0.54–0.83g (mean 0.64± 0.09g) for the purchased honey samples. There was no significant difference (P < 0.05) in the ash content among all the samples (Table 1).

The mineral composition of honey could be influenced by the geographical and botanical origin of the flora, the type and activity of the bees, the extraction technique, and the storage conditions [48]. The results of this study are consistent with the findings of [3], who reported ash contents ranging from 0.21 to 0.57% for stingless bee honey. Although the overall average ash content of stingless bee honey is similar to the purchased honey from the market (0.5 ± 0.00%), the results of the present study suggest that honey produced from the stingless bee (*M. beccarii*) in Ethiopia is richer in mineral content.

This is significant, as the mineral content in honey can have important implications for its for various aspects; used as a marker to identify the geographical origin of the honey, allow for honey traceability and authentication [49]; and indicates the predominant floral sources that the bees have visited, as different plant species accumulate varying levels of minerals in their nectar [50]; used to assess the quality and authenticity of honey, as adulteration or improper processing can alter the natural mineral profile [51]; and it is relevant to the honey's nutritional value, as it provides information about the bioavailability of essential minerals, such as potassium, calcium, and magnesium [52]. It is also established that the ash content of stingless bee honey varies by region, with Southeast Asia ranging from 0.1% to 0.3% [26] and South America between 0.2% and 0.5% [24]. The lower ash contents for *Meliponula beccarii* honey from Ethiopia suggests that Ethiopian honey may have a reduced mineral profile, likely due to the specific flora and soil characteristics in the region and such differences highlight the impact of local biodiversity on honey composition and its potential nutritional value [24–26].

## Sugar contents

The mean sugar concentrations reported were glucose (9.39 ± 4.26 g/100g), sucrose (0.24 ± 0.01 g/100g), maltose (10.81 ± 4.95 g/100g), fructose (16.57 ± 2.55 g/100g), and turanose (0.20 ± 0.00 g/100g). Interestingly, the only significant difference ($p<0.05$) between the two districts was observed for the fructose concentration. Comparing these results to a previous study, the sugar concentrations recorded were generally lower are fructose (36.48 ± 0.54 g/100g), glucose (27.67 ± 0.43 g/100g), and sucrose (1.24 ± 0.18 g/100g) in stingless bee honey samples [36]. The Melipona honeys, typically contain small quantities of maltose [53]. The sugar composition reveals that Southeast Asian honey has fructose levels ranging from 35% to 45% and glucose levels from 25% to 35%. South American honey also shows high fructose levels of 40% to 50%, which aligns closely with the Southeast Asian findings. The balance of these sugars influences the taste, sweetness, and overall quality of the honey [24, 25].

Previous studies has reported maltose levels in stingless bee honey generally range from lower values, typically around 1% to 5% [24, 25]. However, the high maltose content in our stingless bee honey may be due to several factors like specific floral sources can significantly influence sugar composition, with certain plants containing higher maltose concentrations [24], fermentation processes within the nest may also increase maltose levels if honey is stored for extended periods [25], the foraging behavior of stingless bees that could lead to the preferential collection of nectar from maltose-rich plants and environmental factors, such as soil composition and climate, that may further impact floral diversity and nectar quality, resulting in elevated maltose levels [54], processing and storage methods might alter sugar profiles, with prolonged storage or heat affecting composition [26], variability among stingless bee species [25].

These differences in sugar composition could be attributed to various factors, such as the geographical and botanical origins of the honey, the species of stingless bees involved, and the extraction and processing methods employed [29, 49, 51, 52, 55].

### Hydroxymethylfurfural (HMF)

In this study, 0.06 and 4.2 values were recorded as minimum and maximum with an overall mean of 0.89±1.14 mg/kg (Table 1). The highest HMF value was recorded in honey samples collected from the Dandi district (4.20 mg/kg), and the difference in HMF value recorded in honey samples collected from both districts is not significant ($p > 0.05$). This finding disagrees with [3] that stated a high overall HMF range (11.2–22.4mg/kg) and the value reported by [36] (6.58±0.36mg/kg) for pooled honey samples from 14 districts of Oromia, Ethiopia. [21] permits a maximum of 40 mg/kg for the HMF standard. The low HMF content including the purchased honey sample in this study indicates the freshness and good quality of the honey [56–58]. The Hydroxymethylfurfural (HMF) is used as an indicator of poor quality due to adulteration or poor handling like inappropriate storage and the lower the HMF, the freshness and good quality of the honey [9]. HMF levels in Southeast Asian honey are reported at 5 to 10 mg/kg, while in South America, they can range from 10 to 20 mg/kg, indicating Ethiopian honey has an HMF content within the acceptable limits for freshness [25, 54].

### Electrical conductivity

The overall mean calculated value of electrical conductivity of the 24 honey samples of stingless bees from the two study districts was 1.31±0.44 mS/cm, ranging from 0.72–1.99. The electrical conductivity values ranged from 0.72–1.99 mS/cm (mean 1.38±0.48) for the Dandi district samples, 0.75–1.85mS/cm (mean 1.24±0.40) for the Meta Robi district and 0.14–1.50mS/cm (mean 1.79±1.34) for purchased honey (Table 1). The electrical conductivity difference of the two-district sampled honey is not significant ($P > 0.05$), suggesting similarities of flora compositions between the districts. The electrical conductivity in the current study is in line with [3, 36] on stingless bee honey in Ethiopia. The conductivity values of the Melipona honeys vary from 0.32–0.44mS/cm [59]. The electrical conductivity values of the explored honey samples were within the range of allowable value of international standard value (i.e, greater than 0.8 mS/cm) [21]. The significance differences in electric conductivity of honey samples between the two locations suggest its usefulness for differentiating honey of different botanical origins and purity [28].

The unique findings from the data are that the stingless bee (*Meliponula beccarii*) honey from the Dandi and Meta Robi districts of Ethiopia exhibits high acidity, low pH, low HMF, and a distinctive sugar profile, differentiating it from other places studied stingless bee honey [3] indicating potential medicinal properties.

### Conclusion

This study assessed the physicochemical properties of honey produced by the ground-nesting stingless bee (Meliponula beccarii) in Ethiopia and compared these properties to those of stingless bee honey of same species from Ethiopia, stingless bee honey from Southeast Asia, and South America, as well as to the Codex Alimentarius standards for *Apis mellifera* honey and showed both similarities and differences.

Furthermore, the study highlights the distinctive physicochemical profile of stingless bee honey, characterized by significantly higher free acidity (306.64 meq/kg vs. 50 meq/kg), elevated moisture content (28.05% vs. 20%), lower pH (3.29 vs. 3.4–6.1), and higher electrical conductivity (1.31 mS/cm vs. 0.8 mS/cm). Its hydroxymethylfurfural (HMF) content (0.89 mg/kg) is well within Codex standards (40 mg/kg), indicating freshness and quality. The unique low pH and high acidity contribute to potential medicinal benefits. A key finding of this study is the variability between freshly collected and marketed honey samples, with fresh honey consistently demonstrating superior quality indicators compared to marketed counterparts, which

often have higher acidity and moisture levels underscoring the need for specific quality standards and regulations for stingless bee honey to ensure its authenticity and efficacy.

While the promising characteristics of stingless bee honey position it as a valuable natural remedy, the study also acknowledges limitations, including a restricted scope, lack of seasonal variation analysis, absence of bioactive compound profiling, and insufficient comparative studies with other honey types. Addressing these gaps in future research will enhance our understanding of stingless bee honey's full potential for medicinal and nutritional applications, supporting its standardization and utilization across various sectors.

## Recommendations

Based on the findings of this study, establishing specific quality standards and regulations for stingless bee (Meliponula beccarii) honey, given its distinctive physicochemical profile that differs from the Codex Alimentarius standards *for Apis mellifera honey*, and develop certification and labeling schemes to ensure its authenticity and quality; explore the potential medicinal applications of stingless bee honey, leveraging its unique properties such as high acidity, low pH, and low HMF content, and conduct further research to fully characterize its bioactive compounds and therapeutic potential; investigate the potential nutritional benefits. While promoting the diversification of honey sources; support the conservation and sustainable management of stingless bee populations, and develop training and extension programs to educate local communities on their importance; and encourage interdisciplinary collaboration between researchers, beneficiaries, policymakers, and end-users to foster the development and utilization of stingless bee honey as a valuable natural resource.

## Supporting information

**S1 Data.**
(XLSX)

## Acknowledgments

The authors wish to express their gratitude to Ambo University, Mamo Mezemir Campus, Department of Animal Science, for allowing this work to proceed and for its unwavering support throughout the data collection process.

## Author Contributions

**Conceptualization:** Desalegn Begna, Tamiru Kuru.

**Data curation:** Desalegn Begna, Girma Motuma.

**Formal analysis:** Desalegn Begna, Girma Motuma.

**Investigation:** Desalegn Begna.

**Methodology:** Desalegn Begna.

**Project administration:** Shambel Boki, Niguse Bekele.

**Resources:** Shambel Boki, Niguse Bekele, Tamiru Kuru, Achalu Chimdi.

**Writing – original draft:** Desalegn Begna.

**Writing – review & editing:** Desalegn Begna, Girma Motuma, Tamiru Kuru, Achalu Chimdi.

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
