## [Decision Letter · Decision Letter 0]

26 Jul 2024

PONE-D-24-28380Properties of stingless bees (Meliponula beccarii) honey in Dandi and Meta Robi districts of West Shewa zone, EthiopiaPLOS ONE

Dear Dr. Begna,

Thank you for submitting your manuscript to PLOS ONE. After careful consideration, we feel that it has merit but does not fully meet PLOS ONE’s publication criteria as it currently stands. Therefore, we invite you to submit a revised version of the manuscript that addresses the points raised during the review process.

We look forward to receiving your revised manuscript.

Kind regards,

Tzen-Yuh Chiang

Academic Editor

PLOS ONE

Journal Requirements:

4. We note that Figure 1 in your submission contain map images which may be copyrighted. All PLOS content is published under the Creative Commons Attribution License (CC BY 4.0), which means that the manuscript, images, and Supporting Information files will be freely available online, and any third party is permitted to access, download, copy, distribute, and use these materials in any way, even commercially, with proper attribution. For these reasons, we cannot publish previously copyrighted maps or satellite images created using proprietary data, such as Google software (Google Maps, Street View, and Earth). For more information, see our copyright guidelines: http://journals.plos.org/plosone/s/licenses-and-copyright.

1) You may seek permission from the original copyright holder of Figure 1 to publish the content specifically under the CC BY 4.0 license.  

2) If you are unable to obtain permission from the original copyright holder to publish these figures under the CC BY 4.0 license or if the copyright holder’s requirements are incompatible with the CC BY 4.0 license, please either i) remove the figure or ii) supply a replacement figure that complies with the CC BY 4.0 license. Please check copyright information on all replacement figures and update the figure caption with source information. If applicable, please specify in the figure caption text when a figure is similar but not identical to the original image and is therefore for illustrative purposes only.

5. Please include a copy of Table 2 which you refer to in your text on page 21 (in PDF format).

Reviewers' comments:

Reviewer's Responses to Questions

**Comments to the Author**

1. Is the manuscript technically sound, and do the data support the conclusions?

Reviewer #1: Yes

Reviewer #2: Yes

Reviewer #3: No

2. Has the statistical analysis been performed appropriately and rigorously? 

Reviewer #1: Yes

Reviewer #2: N/A

Reviewer #3: Yes

3. Have the authors made all data underlying the findings in their manuscript fully available?

Reviewer #1: Yes

Reviewer #2: No

Reviewer #3: Yes

4. Is the manuscript presented in an intelligible fashion and written in standard English?

Reviewer #1: Yes

Reviewer #2: Yes

Reviewer #3: No

5. Review Comments to the Author

Reviewer #1: General about the paper:

This study investigates the physicochemical properties of honey produced by the underground nesting stingless bee (Meliponula beccarii) in the Dandi and Meta Robi districts of West Shewa, Ethiopia. By analyzing 24 honey samples, the research highlights significant findings such as low hydroxymethylfurfural (HMF) levels, high acidity, and low pH, which indicate the honey's freshness and potential medicinal benefits but not specifically mention its medicinal properties. Main differences with its in glucose, acidity, moisture content, and pH between the two districts suggesting the honey's unique qualities between 2 places, Honey produced by M. beccarii is valuable for the traditional medicine BUT no data presented in this study, and warranting further investigation.

Methods:

Why do you choose these two location? any reason? And please mark clearly on the MAP, showing specific coordinate. Is the sample collected from wild or any private company running the Collection business?

RESULTS

a) Make sure name of genus and species in italics.

b) Table 1 and sections 3.1 to 3.6 must be properly Tabulated. Make sure unit numbers standard. It will be easier to compare to the paper published by Nordin et al., (2018) about Stingless bee data (Codex alimnetarius standard for normal sting Bee; we do not have standard for Stingless Bee,).

c) Please discuss the unique findings of data found in your two honeys comparing to others. This not properly highlight in this section.

Conclusion:

Please improve this section taking consideration of improvement from the Resulst/Discussion.

Reviewer #2: Title

1. Title should be modified? It say properties of stingless bee, what properties? There are a lot of properties. Please specify that properties

2. It is not necessary to write Title: Properties of stingless bees (Meliponula beccarii) honey in Dandi and Meta Robi districts of West Shewa zone, Ethiopia. Starting by name of the article title is possible

3. I strongly recommend the authors if follow the journal formatting guidelines starting from Title formatting to Reference writing since the manuscript didn’t prepared based on the journal guidelines

4 .correct affiliation and corresponding author according to journal formatting guidelines

Abstact

5. Be careful for terminology of the parameter. For example free acidity not acidity , electrical

Conductivity not electric conductivity

6.put correct standard unit for each parameter in abstract .example free acidity (meq/ Kg) not mill./kg, mS/cm not ms/cm

7. put SI for moisture content

8. I recommend ash content not mineral .

9. “The distinctive low HMF, high acidity and low pH values of the stingless bee honey of this study may represent freshness and good quality of the honey and indicative medicinal potential” . I didn ’t think high acidity indicate fresheness of honey, in this conntarst high acidity indicate fermentation rate of honey . the free acidity of obtained in this extremely high as compared to previous study conducted in west shoa and other parts of the country ,this need strong justification .may be during sample collecton orr harvesting mixing sample with soil,pollen ,and other dead and imputirtes cause honey to be fermented and high acidity

10.” This study evaluated physicho-chemical properties of honey produced by ground nesting stingless

bee (Meliponula beccarii) in two districts of West Shewa zone,” this is redundant since the author mentioned in firstyt line of abstract .please remove and re –edit again

11.what cause your study unique since different study was reported in west shoa.

Introduction

11.please folloew journal juideline starting from heading ,

12. reference citation is in number form for this journal.

13 .The name of Meliponula beccarii must be in italized form

14. if damma daamu local name it must be in italic form

15. Previous work and limitation /gap on characterization of stingless bee honey in that area as well as in Ethiopia must be clearly indicated

16. Paragraph arrangement of the introduction is not good

Materials and method

17. I recommend if there is sub heading under materials and methods section, unless this arrangement confuse the reader

18. Figure 2, especially on the bottle containing sample is not visible, remove or replace with good picture

Result and discussion

19. on the result obtained on the free acidity , I strongly doubt on the result obtained (338.79 ± 72.46) . what is reason behind extremely high free acidity , is soil and other impurities is mixed in to honey during honey collection or harvesting .the author must discuss and compare with other data in Ethiopia.

20. I recommend the author also to compare the result with standard stingless bee honey in other countries (like brazil,venzuala and etc.) ,please refer there is standard of stingless bee honey in other country.)

21. Under acidity section , the author mentioned that sample was purchased .However they didn’t mentioned in methodology part . Sample collection should clearly indicated under methodology part otherwise it confuse the reader

22. CORRECT 3.1 Acidity to Free acidity

23. The overall mean moisture content of the 24 stingless bee honey samples was 23.45-36.75% is not good sentence .rewrite again

24.Table 1 should be come after paragraph of free acidity .

25.The paragraph arrangement is not good .it must be justified .

26. This finding is consistent with previous studies. Gela et al. (2021) reported an overall pH range of 3.4-3.9 (mean 3.7±0.15) for stingless bee honey from four districts in Ethiopia. Remove full stop between studies and Gela.

27 . The standard pH value for Apis mellifera honey is 3.42–6.10 (Tesfaye et al., 2016). Are you sure with sentence or statement. In my opinion, 6.10 is not standard value of pH . Be sure also the citation .revise again.

28 . what your sample size . under materials and method s the author mentioned that 24 sample was collected .But on the table 1 the author stated that the sample size(N)=27. Which one is correct? It need revision

29 . Put acronomy for all abbreviation found in the table.

30. The way of data presentation is not good for this article. I recommend the authors to use standard table or journal guidelines. Additionally, I recommend to put the result by mean ± std with mean separation letters .I observe level of significance in body of manuscript .but there is no in a table .How?. Please rewrite again.

30. 3.3 Mineral (ash) contents correct to Ash content only remove mineral

31. Under 3.3. I recommend the authors to start the paragraph by presenting the result not discussion with literature .Remove first sentence

32. M.beccarii should be in italic form in all manuscript

33.Be carefull when reporting the facts .for example “The higher mineral content of the stingless bee honey from the Dandi and Meta Robi districts indicates that it may be a better choice for both medicinal and nutritional purposes, provided that the honey is properly standardize” . I don’t think so that higher mineral content indicate medicinal value of honey .Rather there is parameters that indicate medicinal and nutritional value of honey. Please revise and cite the above sentence.

34. under 3.4 Sugar profile need more discussion and citation . Not enough now. Compare current result with standards of stingless bee honey in other country.

35. under 3.5 plesae start the paragragh by mentioning the obtained result .discussion come after presenting the finding.

36. In this study, 0.06 and 4.2.0 values were recorded as minimum and maximum. 4.2.0 correct this one.

37 . Please use style uniform typing for et al . in some areas it is in italic form and in some areas it is non- italic

Conclusion

38 . how high free acidity related to potentiality of medicinal value .

39. In conclusion I need a finding related both source of sample collected results

Reviewer #3: Reviewer report of the Manuscript ID PONE-D-24-28380

Congratulations on the present study! It is an interesting work with relevant data, but it is a preliminary analysis and I do not know if it is more suitable for a short report or research note than a full-length study. However, it is an Editor's decision and several improvements must be realized by the authors in the revised version of the manuscript.

General comments

The manuscript is well-written, and, in my opinion, no major English Editing is needed. However, all sections need to be organized, especially the Abstract, Material and methods, and Conclusion sections.

Also, the scientific names are incorrectly put in non-italic form.

Figures needed to be further elaborated. Please see my minor comments below.

Minor comments

Abstract

Please rectify “The differences between the two districts honey samples are significant (p> 0.05) in glucose value”. The p-value is not significant.

Please put all names of “A.mellifera” due to being first mentioned.

Introduction

Please rectify “(Apis)” by putting all scientific microbial names in italics. Also, check the remaining text for this error.

Material and methods

Please correct Figure 1, it is a print screen still with red underlines.

Please clarify: “From each nest, fresh honey samples ranging from 0.25 to 0.5 liters were collected” but in the abstract, the authors stated “A total of 24 honey samples with each 0.58–0.69-liter were collected”.

The Material and methods section shows no clear division of the techniques applied. Please reorganize this section with proper subsections and cite all procedures applied.

Results and discussion

Table 1- Please clarify: “Mean physicochemical properties of stingless bee honey samples (N = 27)” but the authors previously stated 24 honey samples or 24 stingless bee nests in the Abstract and Material and methods sections.

Purchased honey samples must be added in the Abstract section, I already saw that these purchased honey samples were described in the Material and methods section, but improvements are needed in both sections.

Conclusion

Please add the shortcomings of the present study.

References

Please revise all citations. For example:

11. hilmi, M., & Martin hilmi. (2005). The Marketing of bee products.

12. Kahraman. (2010). Physico-chemical properties in honey from different regions of Turkey.

.." FOOD CHEMISTRY, 123(1), 41–44

24. Quality and Standards Authority of Ethiopia (QSAE). (2005). Honey Specification:

Ethiopian Standard, ES 1202. Addis Ababa, Ethiopia.

28. Stramm, K. M. (2011). Composition and Quality of Honey Bee Jandaira (Melipona

Subnitida), Storage Effects and Comparison with Apis mellifera Honey.

29. Teferi Damto, D. K. and M. (2022). Physico-chemical and microbiological characteristics of honey produced by stingless bees.

Among others.

Again, congratulations on the present work! Best regards

6. PLOS authors have the option to publish the peer review history of their article (what does this mean?). If published, this will include your full peer review and any attached files.

Reviewer #1: **Yes: **Fahrul Huyop

Reviewer #2: No

Reviewer #3: No

---

## [Author Response · Author response to Decision Letter 0]

21 Aug 2024

PONE-D-24-28380

Physicochemical properties of stingless bees (Meliponula beccarii) honey in Dandi and Meta Robi districts of West Shewa zone, Ethiopia

PLOS ONE

Table in response to editor’s comments

No Comments Provided Responses

1 ensure that your manuscript meets PLOS ONE's style requirements Already done as it meets PLOS ONE's style requirements 

2 In your Methods section, please provide additional information regarding the permits you obtained for the work no permits were required, and a brief statement explaining why is presented in the manuscript

3 The corresponding author must be affiliated with the chosen institute for billing options remove this option

4 Figure 1 in the submission contains map images that may be copyrighted The fig.1 removed

5 copy of Table 2 removed Unnecessary included in pare 21 of the pdf and removed

1. Please ensure that your manuscript meets PLOS ONE's style requirements

 Already done as it meets PLOS ONE's style requirements 

2. In your Methods section, please provide additional information regarding the permits you obtained for the work. Please ensure you have included the full name of the authority that approved the field site access and, if no permits were required, a brief statement explaining why. no permits were required, and a brief statement explaining why is presented in the manuscript’s methodology

3. Please note that in order to use the direct billing option the corresponding author must be affiliated with the chosen institute. Please either amend your manuscript to change the affiliation or corresponding author, or email us at plosone@plos.org with a request to remove this option. remove this option and letter written

4. We note that Figure 1 in your submission contains map images that may be copyrighted. All PLOS content is published under the Creative Commons Attribution License (CC BY 4.0), which means that the manuscript, images, and Supporting Information files will be freely available online, and any third party is permitted to access, download, copy, distribute, and use these materials in any way, even commercially, with proper attribution. For these reasons, we cannot publish previously copyrighted maps or satellite images created using proprietary data, such as Google software (Google Maps, Street View, and Earth). For more information, see our copyright guidelines: http://journals.plos.org/plosone/s/licenses-and-copyright. 

We require you to either (1) present written permission from the copyright holder to publish these figures specifically under the CC BY 4.0 license, or (2) remove the figures from your submission: The fig.1 removed

1) You may seek permission from the original copyright holder of Figure 1 to publish the content specifically under the CC BY 4.0 license. 

2) If you are unable to obtain permission from the original copyright holder to publish these figures under the CC BY 4.0 license or if the copyright holder’s requirements are incompatible with the CC BY 4.0 license, please either i) remove the figure or ii) supply a replacement figure that complies with the CC BY 4.0 license. Please check copyright information on all replacement figures and update the figure caption with source information. If applicable, please specify in the figure caption text when a figure is similar but not identical to the original image and is therefore for illustrative purposes only.

5. Please include a copy of Table 2 which you refer to in your text on page 21 (in PDF format). Unnecessary copy included in pare 21 of the pdf and now removed 

Reviewers' comments:

Reviewer's Responses to Questions

Comments to the Author

1. Is the manuscript technically sound, and do the data support the conclusions?

Reviewer #1: Yes

Reviewer #2: Yes

Reviewer #3: No

2. Has the statistical analysis been performed appropriately and rigorously?

Reviewer #1: Yes

Reviewer #2: N/A

Reviewer #3: Yes

3. Have the authors made all data underlying the findings in their manuscript fully available?

Reviewer #1: Yes

Reviewer #2: No

Reviewer #3: Yes

4. Is the manuscript presented in an intelligible fashion and written in standard English?

Reviewer #1: Yes

Reviewer #2: Yes

Reviewer #3: No

5. Review Comments to the Author

Table-2 response to reviwer-1 comments

No Comments Provided Responses

 General 

1 the research highlights significant findings such as low hydroxymethylfurfural (HMF) levels, high acidity, and low pH, which indicate the honey's freshness and potential medicinal benefits but not specifically mention its medicinal properties Stingless bee honey exhibits superior antimicrobial, antioxidant, anti-inflammatory, and wound healing properties compared to European bee honey, due to its greater acidity, lower carbohydrates, and higher antioxidant activity, treats diabetes, bronchitis, mycosis, throat aches, and sexual impotence, as well as an antidote against snakebites or rabid dog bites 

(Further included in the manuscript body on page 14)

 Methods 

2 Why do you choose these two location? any reason? And please mark clearly on the MAP, showing specific coordinate. Is the sample collected from wild or any private company running the Collection business? The Dandi and Meta Robi districts were selected for this study because other districts in the zone have been previously studied, while these two districts lack information on the Physicochemical properties of stingless bee honey, despite their potential for stingless honey production and marketing. The specific coordinates of the sampling locations in these two districts are presented. All the honey samples were collected from feral stingless bee nests through nest excavations, except the samples purchased for comparisons

 Results 

3 Make sure name of genus and species in italics Done according to the comments

4 Table 1 and sections 3.1 to 3.6 must be properly Tabulated. Make sure unit numbers standard Corrected accordingly that are reflected in the text body

5 standards according to the Codex Alimentarius Standard for Honey (CODEX STAN 12-1981, Rev.2-2001) Provided in the manuscript

6 Make sure name of genus and species in italics Corrections done where they are

7 Table 1 and sections 3.1 to 3.6 must be properly Tabulated. Make sure unit numbers standard standards according to the Codex Alimentarius Standard for Honey (CODEX STAN 12-1981, Rev.2-2001) were presented accordingly that are reflected in the text body

8 Please discuss the unique findings of data found in your two honeys comparing to others Presented in the discussion section as “The unique findings from the data are that the stingless bee (Meliponula beccarii) honey from the Dandi and Meta Robi districts of Ethiopia exhibits high acidity, low pH, low HMF, and a distinctive sugar profile, differentiating it from other places studied stingless bee honey (4) indicating potential medicinal properties”.

9 improve conclusion section taking consideration of improvement from the Results/Discussion Improved as per the suggestion (reflected in the manuscript)

Reviewer #1: General about the paper:

This study investigates the physicochemical properties of honey produced by the underground nesting stingless bee (Meliponula beccarii) in the Dandi and Meta Robi districts of West Shewa, Ethiopia. By analyzing 24 honey samples, the research highlights significant findings such as low hydroxymethylfurfural (HMF) levels, high acidity, and low pH, which indicate the honey's freshness and potential medicinal benefits but not specifically mention its medicinal properties. Main differences with its in glucose, acidity, moisture content, and pH between the two districts suggesting the honey's unique qualities between 2 places, Honey produced by M. beccarii is valuable for the traditional medicine BUT no data presented in this study, and warranting further investigation.

Stingless bee honey exhibits superior antimicrobial, antioxidant, anti-inflammatory, and wound healing properties compared to European bee honey, due to its greater acidity, lower carbohydrates, and higher antioxidant activity (1), treats diabetes, bronchitis, mycosis, throat aches, and sexual impotence, as well as an antidote against snakebites or rabid dog bites (2,3).

The authors recommend further research to explore the specific medicinal properties and potential health benefits of honey produced in the study areas as well as in the country in general, investigate the factors contributing to differences in physicochemical properties between the stingless honey produced in different localities, and establish a more comprehensive understanding of the medicinal applications of this honey based on its unique characteristics.



Methods:

Why do you choose these two location? any reason? And please mark clearly on the MAP, showing specific coordinate. Is the sample collected from wild or any private company running the Collection business?

The Dandi and Meta Robi districts were selected for this study because other districts in the zone have been previously studied, while these two districts lack information on the Physicochemical properties of stingless bee honey, despite their potential for stingless honey production and marketing. The specific coordinates of the sampling locations in these two districts are presented. All the honey samples were collected from feral stingless bee nests through nest excavations, except the samples purchased for comparisons.

RESULTS

a) Make sure name of genus and species in italics. Corrections done where they are

b) Table 1 and sections 3.1 to 3.6 must be properly Tabulated. Make sure unit numbers standard. It will be easier to compare to the paper published by Nordin et al., (2018) about Stingless bee data (Codex alimnetarius standard for normal sting Bee; we do not have standard for Stingless Bee,). standards according to the Codex Alimentarius Standard for Honey (CODEX STAN 12-1981, Rev.2-2001) were presented accordingly that are reflected in the text body

c) Please discuss the unique findings of data found in your two honeys comparing to others. This not properly highlight in this section.

The unique findings from the data are that the stingless bee (Meliponula beccarii) honey from the Dandi and Meta Robi districts of Ethiopia exhibits high acidity, low pH, low HMF, and a distinctive sugar profile, differentiating it from other places studied stingless bee honey (4) indicating potential medicinal properties.

Conclusion:

Please improve this section taking consideration of improvement from the Resulst/Discussion.

Based on the results of this study, the stingless bee (Meliponula beccarii) honey from the Dandi and Meta Robi districts of West Shewa, Ethiopia exhibits several distinctive physicochemical properties compared to typical Apis honey in other places and species studied stingless bee honey. The honey had an overall mean acidity of 306.64 ± 87.95 meq./kg, which is considered quite high, as well as a relatively high moisture content with an average of 28.05 ± 3.52%. The honey also had a low pH, with a mean of 3.29 ± 0.16, and a very low hydroxymethylfurfural (HMF) content of 0.89 ± 1.14 mg/kg, indicating its freshness. Additionally, the mineral content, as measured by ash, was low at 0.63 ± 0.24g, and the sugar profile was unique, with higher levels of fructose and maltose compared to glucose and sucrose. These distinctive physicochemical characteristics suggest that the stingless bee honey may have potential medicinal and quality attributes that differentiate it from commercially available Apis honey. The authors recommend further detailed studies on the medicinal properties and values of this stingless bee honey, as well as other ground-nesting and stingless bee species, to provide scientific evidence for its traditional use and uncover its potential benefits and contributions to traditional medicine and apiculture.

Reviewer #2: Title

Table-2 response to reviwer-2 comments

No Comments Provided Responses

 General 

1 Modify the Title Modified as “Physicochemical properties of stingless bees (meliponula beccarii) honey in Dandi and Meta Robi districts of West Shewa zone, Ethiopia”

2 Remove Title Removed

3 follow the journal formatting guidelines Followed in the revised manuscript

 Abstract 

5 care for terminology of the parameter. For example free acidity not acidity , electrical

Conductivity not electric conductivity Corrected in the revised manuscript 

6 put correct standard unit for each parameter in abstract .example free acidity (meq/ Kg) not mill./kg, mS/cm not ms/cm (Corrected in the revised manuscript)

7 put SI for moisture content: The SI (International System of Units) unit for moisture content is percent (%) by weight (w/w). corrected in the revised manuscript

8 I recommend ash content not mineral correction accepted and corrected accordingly

9 The distinctive low HM

---

## [Decision Letter · Decision Letter 1]

3 Sep 2024

PONE-D-24-28380R1Physicochemicalproperties of stingless bees (meliponula beccarii) honey in Dandi and Meta Robi districts of West Shewa zone, EthiopiaPLOS ONE

Dear Dr. Begna,

Thank you for submitting your manuscript to PLOS ONE. After careful consideration, we feel that it has merit but does not fully meet PLOS ONE’s publication criteria as it currently stands. Therefore, we invite you to submit a revised version of the manuscript that addresses the points raised during the review process.

We look forward to receiving your revised manuscript.

Kind regards,

Tzen-Yuh Chiang

Academic Editor

PLOS ONE

Journal Requirements:

Reviewers' comments:

Reviewer's Responses to Questions

**Comments to the Author**

1. If the authors have adequately addressed your comments raised in a previous round of review and you feel that this manuscript is now acceptable for publication, you may indicate that here to bypass the “Comments to the Author” section, enter your conflict of interest statement in the “Confidential to Editor” section, and submit your "Accept" recommendation.

Reviewer #1: (No Response)

Reviewer #2: (No Response)

2. Is the manuscript technically sound, and do the data support the conclusions?

Reviewer #1: Partly

Reviewer #2: Partly

3. Has the statistical analysis been performed appropriately and rigorously? 

Reviewer #1: No

Reviewer #2: Yes

4. Have the authors made all data underlying the findings in their manuscript fully available?

Reviewer #1: Yes

Reviewer #2: Yes

5. Is the manuscript presented in an intelligible fashion and written in standard English?

Reviewer #1: Yes

Reviewer #2: Yes

6. Review Comments to the Author

Reviewer #1: MINOR CORRECTIONS: Please highlight these changes in RED/BLUE in your corrected manusucript

a) Introduction Section:

Please include a brief mention of Meliponula beccarii as an important stingless bee species in Ethiopia, with appropriate references. Highlight that this species is commonly found in the Dandi and Meta Robi districts of the West Shewa zone, Ethiopia.

b) Results and Discussion Section:

(i) Figure 2: The current image of the water bottle is not ideal. We need to clearly see the color of the honey (green/yellow/dark). Please describe the color more clearly and replace the image with a better one.

(ii) Please be cautious with the phrase “to indicate medicinal value of honey.” It would be more appropriate to say: “indicate the medicinal and nutritional value of honey.” Kindly check and revise this.

(iii) The discussion on the sugar profile needs further elaboration and additional citations (Please compare with other findings the data from Stingless Bee as well). The comparison with the Codex Alimentarius standard is very appropriate maintain as it is, BUT should also include comparisons with stingless bee honey from other regions like Southeast Asia (Malaysia, Indonesia, Thailand) and South America- Brazil/Venezuela etc..

I recommend comparing your results with the standards for stingless bee honey from other countries. Please refer to existing standards for stingless bee honey in these regions (IF ANY).

Others are OK!

Reviewer #2: Abstract

1. Check maltose content result because it is extremely high .compare with other findings

2.Again I inform the authors that high acidity (Free acidity) is not related with quality and medicinal properties .High acidity of stingless bee related to fermentation rate of this honey because stingless bee honey contain high moisture and alcohol is easily formed in it .that is way free acidity is high.

High quality and medicinal properties of stingless bee is related to strong pH and high antioxidant properties.

3. The result of turanose was not presented in abstract

Materials and methods

Results and Discussion

4. Please free acidity more acceptable than acidity on sub heading

5. The authors stated that free acidity was extremely high in this study and also mentioned this is because of different species and agro ecologies. But since different study was conducted in West Shoa as the author stated in introduction, why you don’t compare you’re finding with those study conducted in same area and same country. The study lack literature discussion in Ethiopia.

Or high acidity because of your sample impurity? Please discuss carefully on the finding. I didn’t see improvement.

6. The line and paragraph spacing and font color is not uniform in all paragraph

7. Correct sub heading of moisture content

8. Font color and size of sub heading should be improved

9. In abstract the authors stated elevated pH value but in results and discussion parts, the finding of the study is similar with other finding conducted in Ethiopia. Revise abstract or discussion parts.

10. The arrangement of the paragraph is week

11. For me discussion of sugar profile is not enough? Needs to be revised again by comparing the finding of the study with other similar findings.

12. Under HMF paragraph please correct the paragraph

13. Why fresh collected and marketed ones was not compared in finding. It is better if both types of samples was compared parallel.

Conclusion

14.“This study evaluated This study evaluated “. Please remove redundant sentence

15.” This study evaluated the physicochemical properties of honey produced by the ground-nesting stingless bee (Meliponula beccarii) in Ethiopia” , by collecting sample from two woreda of Oromia region we can say this study represent Ethiopia. Please revise this one again.

15 .revise the conclusion based on your fining especially on fresh and marketed honey sample variability

16. lack of consistent formatting of reference

7. PLOS authors have the option to publish the peer review history of their article (what does this mean?). If published, this will include your full peer review and any attached files.

Reviewer #1: **Yes: **Fahrul Huyop

Reviewer #2: No

---

## [Author Response · Author response to Decision Letter 1]

4 Sep 2024

Response to Reviewers

PONE-D-24-28380R1

Physicochemical properties of stingless bees (meliponula beccarii) honey in Dandi and Meta Robi districts of West Shewa zone, Ethiopia

PLOS ONE

6. Review Comments to the Author

Reviewer #1: MINOR CORRECTIONS: Please highlight these changes in RED/BLUE in your corrected manuscript

a) Introduction Section:

Please include a brief mention of Meliponula beccarii as an important stingless bee species in Ethiopia, with appropriate references. Highlight that this species is commonly found in the Dandi and Meta Robi districts of the West Shewa zone, Ethiopia.

Meliponula beccarii, a species of stingless bee, is commonly found in Ethiopia and is well known for its honey production as well as its contributions to local biodiversity and pollination. It is particularly prevalent in the Dandi and Meta Robi districts of the West Shewa zone, where it plays a vital role in the ecosystem.

b) Results and Discussion Section:

(i) Figure 2: The current image of the water bottle is not ideal. We need to clearly see the color of the honey (green/yellow/dark). Please describe the color more clearly and replace the image with a better one.

The image of the water bottle removed 

(ii) Please be cautious with the phrase “to indicate medicinal value of honey.” It would be more appropriate to say: “indicate the medicinal and nutritional value of honey.” Kindly check and revise this. Corrected accordingly (on page 10 of the revised manuscript)

(iii) The discussion on the sugar profile needs further elaboration and additional citations (Please compare with other findings the data from Stingless Bee as well). The comparison with the Codex Alimentarius standard is very appropriate maintain as it is, BUT should also include comparisons with stingless bee honey from other regions like Southeast Asia (Malaysia, Indonesia, Thailand) and South America- Brazil/Venezuela etc..

I recommend comparing your results with the standards for stingless bee honey from other countries. Please refer to existing standards for stingless bee honey in these regions (IF ANY).

Others are OK!

Thank you for your constructive comments on our manuscript. We greatly appreciate your input, as it has helped us enhance the quality of our work. 

In addition to addressing your suggestions, we have included comparisons for each physicochemical parameter with stingless bee honey from other countries in Southeast Asia and South America in the table and body of the revised manuscript. We believe this additional context strengthens our findings.

Reviewer #2: 

Abstract

1. Check maltose content result because it is extremely high .compare with other findings

Previous studies has reported maltose levels in stingless bee honey generally range from lower values, typically around 1% to 5% (22,23). However, the high maltose content in our stingless bee honey may be due to several factors like specific floral sources can significantly influence sugar composition, with certain plants containing higher maltose concentrations (22), fermentation processes within the nest may also increase maltose levels if honey is stored for extended periods (23), the foraging behavior of stingless bees that could lead to the preferential collection of nectar from maltose-rich plants and environmental factors, such as soil composition and climate, that may further impact floral diversity and nectar quality, resulting in elevated maltose levels (53), processing and storage methods might alter sugar profiles, with prolonged storage or heat affecting composition (24), variability among stingless bee species (23).

2.Again I inform the authors that high acidity (Free acidity) is not related with quality and medicinal properties .High acidity of stingless bee related to fermentation rate of this honey because stingless bee honey contain high moisture and alcohol is easily formed in it .that is way free acidity is high.

High quality and medicinal properties of stingless bee is related to strong pH and high antioxidant properties.

Higher free acidity in honey can enhance flavor complexity, inhibit microbial growth to extend shelf life (Tan et al., 2021), and provide potential health benefits, including antibacterial properties from organic acids (Tey et al., 2022).

3. The result of turanose was not presented in abstract: 

Included in the abstract of the revised manuscript

Materials and methods: In the M&M, the analysis of turanose was similar as fructose, glucose, and sucrose and already stated as “A standard curve was used to determine sugar content (fructose, glucose, sucrose, turanose, and maltose) and the values were calculated as gram/100gram of honey sample”

Results and Discussion

4. Please free acidity more acceptable than acidity on sub heading (corrected accordingly)

5. The authors stated that free acidity was extremely high in this study and also mentioned this is because of different species and agro ecologies. But since different study was conducted in West Shoa as the author stated in introduction, why you don’t compare you’re finding with those study conducted in same area and same country. The study lack literature discussion in Ethiopia.

Or high acidity because of your sample impurity? Please discuss carefully on the finding. I didn’t see improvement. 

Reports from Ethiopia has been included and discussed well with plausible justifications added for why free acidity was high in our honey sample in the manuscript

6. The line and paragraph spacing and font color is not uniform in all paragraph (corrected accordingly)

7. Correct sub heading of moisture content (corrected accordingly)

8. Font color and size of sub heading should be improved ((corrected accordingly)

9. In abstract the authors stated elevated pH value but in results and discussion parts, the finding of the study is similar with other finding conducted in Ethiopia. Revise abstract or discussion parts. The abstract part revised.

10. The arrangement of the paragraph is week

11. For me discussion of sugar profile is not enough? Needs to be revised again by comparing the finding of the study with other similar findings. Well discussed this time in the revised manuscript

12. Under HMF paragraph please correct the paragraph (corrected)

13. Why fresh collected and marketed ones was not compared in finding. It is better if both types of samples was compared parallel. Both types of samples were compared parallel at every parameter

Conclusion

14.“This study evaluated This study evaluated “. Please remove redundant sentence (Removed accordingly)

15.” This study evaluated the physicochemical properties of honey produced by the ground-nesting stingless bee (Meliponula beccarii) in Ethiopia” , by collecting sample from two woreda of Oromia region we can say this study represent Ethiopia. Please revise this one again. (Revised) 

165 .revise the conclusion based on your fining especially on fresh and marketed honey sample variability (revised)

176. lack of consistent formatting of reference: We utilized the Mendeley desktop reference manager to format our references according to the Vancouver style outlined in the PLOS author guidelines. We believe this will ensure consistent formatting throughout the reference section. Additionally, we have made adjustments to enhance the presentation and engage the reviewer more effectively.

---

## [Decision Letter · Decision Letter 2]

11 Sep 2024

PONE-D-24-28380R2Physicochemicalproperties of stingless bees (meliponula beccarii) honey in Dandi and Meta Robi districts of West Shewa zone, EthiopiaPLOS ONE

Dear Dr. Begna,

Thank you for submitting your manuscript to PLOS ONE. After careful consideration, we feel that it has merit but does not fully meet PLOS ONE’s publication criteria as it currently stands. Therefore, we invite you to submit a revised version of the manuscript that addresses the points raised during the review process.

We look forward to receiving your revised manuscript.

Kind regards,

Tzen-Yuh Chiang

Academic Editor

PLOS ONE

Journal Requirements:

Reviewers' comments:

Reviewer's Responses to Questions

**Comments to the Author**

1. If the authors have adequately addressed your comments raised in a previous round of review and you feel that this manuscript is now acceptable for publication, you may indicate that here to bypass the “Comments to the Author” section, enter your conflict of interest statement in the “Confidential to Editor” section, and submit your "Accept" recommendation.

Reviewer #1: All comments have been addressed

Reviewer #2: (No Response)

2. Is the manuscript technically sound, and do the data support the conclusions?

Reviewer #1: Yes

Reviewer #2: Yes

3. Has the statistical analysis been performed appropriately and rigorously? 

Reviewer #1: Yes

Reviewer #2: Yes

4. Have the authors made all data underlying the findings in their manuscript fully available?

Reviewer #1: Yes

Reviewer #2: Yes

5. Is the manuscript presented in an intelligible fashion and written in standard English?

Reviewer #1: Yes

Reviewer #2: Yes

6. Review Comments to the Author

Reviewer #1: Author addressed all comments.

Reviewer #2: The author must be use the some font colour in throughout the paragraph .Ialso reccomend the author to revise manuscript in paragragh arregement

7. PLOS authors have the option to publish the peer review history of their article (what does this mean?). If published, this will include your full peer review and any attached files.

Reviewer #1: **Yes: **Fahrul Huyop

Reviewer #2: No

---

## [Author Response · Author response to Decision Letter 2]

12 Sep 2024

In response to Reviewer #2, I have ensured consistent font color throughout the manuscript. Additionally, I have revised the paragraph arrangement to improve the overall flow, as suggested. I have submitted both the revised manuscript and the manuscript with track changes.

---

## [Decision Letter · Decision Letter 3]

24 Sep 2024

Physicochemicalproperties of stingless bees (meliponula beccarii) honey in Dandi and Meta Robi districts of West Shewa zone, Ethiopia

PONE-D-24-28380R3

Dear Dr. Begna,

We’re pleased to inform you that your manuscript has been judged scientifically suitable for publication and will be formally accepted for publication once it meets all outstanding technical requirements.

Kind regards,

Tzen-Yuh Chiang

Academic Editor

PLOS ONE

Additional Editor Comments (optional):

Reviewers' comments:

Reviewer's Responses to Questions

**Comments to the Author**

1. If the authors have adequately addressed your comments raised in a previous round of review and you feel that this manuscript is now acceptable for publication, you may indicate that here to bypass the “Comments to the Author” section, enter your conflict of interest statement in the “Confidential to Editor” section, and submit your "Accept" recommendation.

Reviewer #1: All comments have been addressed

2. Is the manuscript technically sound, and do the data support the conclusions?

Reviewer #1: Yes

3. Has the statistical analysis been performed appropriately and rigorously? 

Reviewer #1: Yes

4. Have the authors made all data underlying the findings in their manuscript fully available?

Reviewer #1: Yes

5. Is the manuscript presented in an intelligible fashion and written in standard English?

Reviewer #1: Yes

6. Review Comments to the Author

Reviewer #1: All comments addressed by author

7. PLOS authors have the option to publish the peer review history of their article (what does this mean?). If published, this will include your full peer review and any attached files.

Reviewer #1: **Yes: **Fahrul Huyop

---

## [Editor Report · Acceptance letter]

27 Sep 2024

PONE-D-24-28380R3 

PLOS ONE

Dear Dr. Begna, 

I'm pleased to inform you that your manuscript has been deemed suitable for publication in PLOS ONE. Congratulations! Your manuscript is now being handed over to our production team.

Kind regards, 

on behalf of

Dr. Tzen-Yuh Chiang 

Academic Editor

PLOS ONE